# Microbial and geo-archaeological records reveal the growth rate, origin and composition of desert rock surface communities

Nimrod Wieler[1], Tali Erickson Gini[2], Osnat Gillor[1, *], and Roey Angel[3, *]

[1]Zuckerberg Institute for Water Research, Blaustein Institutes for Desert Research, Ben Gurion University of the Negev, Sede Boqer Campus, Israel
[2]Southern Region, Israel Antiquities Authority, Omer 84965, Israel
[3]Soil and Water Research Infrastructure and Institute of Soil Biology, Biology Centre CAS, České Budějovice, Czechia
[*]These authors contributed equally to this work.

**Correspondence:** Roey Angel (roey.angel@bc.cas.cz) and Osnat Gillor (gilloro@bgu.ac.il)

**Abstract.** Biological rock crusts (BRCs) are ubiquitous features of rock surfaces in drylands composed of slow-growing microbial assemblages. BRC presence is often correlated with rock weathering, soiling effect, or mitigating geomorphic processes. However, their development rate is still unknown. In this work, we characterised and dated BRCs in an arid environment, under natural conditions, by integrating archaeological, microbiological and geological methods. To this end, we sampled rocks from a well-documented Byzantine archaeological site and the surrounding area located in the Central Negev Desert, Israel. The archaeological site, which is dated to the 4th-7th centuries CE, was constructed from two lithologies, limestone and chalk. BRC started developing on the rocks after being carved, and its age should match that of the site. Using stable carbon and oxygen isotope ratios, we confirmed the biogenic nature of the crusts. The BRC samples showed mild differences in the microbial community assemblages between the site and its surrounding, irrespective of lithology, confirming the dominance of aeolian inoculation sources. All BRCs were dominated by Actinobacteria, Cyanobacteria and Proteobacteria. We further measured the BRC thickness on 1700 years old building stone blocks and determined it to be 0.1-0.6 mm thick. Therefore, a BRC growth rate was estimated, for the first time, to be 0.06-0.35 mm 1000 yr$^{-1}$. Our dating method was then validated on a similar archaeological site located ca. 20 km away, giving comparable values. We propose that BRC growth rates could be used as an affordable yet robust dating tool in archaeological sites in arid environments.

## 1  Introduction

In arid and hyper-arid environments where abiotic processes are considered as the primary contributor to landform formation, barren rock surfaces, free of vegetation, are a ubiquitous feature (Owen et al., 2011). These surfaces are exposed to multiple stress factors, such as lack of water, high radiation, and extreme temperature fluctuations, and therefore represent the edge of biotic existence on Earth (Viles, 2008). Such rock surfaces serve as a habitat for microorganisms by providing colonisation strata either on the outside (epilith; Pointing and Belnap, 2012), in the inner pores (endolith; Büdel and Wessels, 1991; Friedmann and Kibler, 1980) or underneath the rock (hypolith; Wierzchos et al., 2013). The microorganisms colonising rocks form a hardy biomineral rock coatings known as a biological rock crust (BRC; Gorbushina, 2007), which is common in most arid

and hyper-arid regions worldwide (Gorbushina, 2007; Lebre et al., 2017; Pointing and Belnap, 2012). BRCs structure, when developed on carbonate-rich minerals (e.g., limestone), often consists of a hardpan-laminated structure composed of masses of micritic to microsparitic carbonate layers interbedded with microbial coatings. Such laminated structures were previously reported by Alonso-Zarza and Wright (2010) as terrestrial calcretes. From a microbial perspective, BRCs are typically colonised by Actinobacteria, Cyanobacteria and other phototrophic and heterotrophic bacteria, yet they lack, or nearly lack, archaea, fungi, or algae (Lang-Yona et al., 2018). BRC inoculum has been proposed to originate from settled dust (Viles, 2008) or the surrounding soil (Makhalanyane et al., 2015). Wieler et al., 2019 noted that the microbial composition of the BRC was distinct from the soil communities but shared some of the communities found in the settled dust. Local contribution of aeolian material (Crouvi et al.) may also affect the microbial communities found in the settled dust. The influence of biota on landscapes is a topic that has remained mostly unexplored (Dietrich and Perron, 2006; Viles, 2019). BRCs were noted to play a crucial role in the functioning of arid and hyper-arid environments because of the limited activity of plants and soil (Pointing and Belnap, 2012). A range of geobiological roles was suggested for BRCs, including rock weathering (Garcia-Pichel, 2006; Warscheid and Braams, 2000), soiling effect (Viles and Gorbushina, 2003), deterioration of architectural heritage sites (Cabello-Briones and Viles, 2017) and mitigating geomorphic processes (Mcllroy de la Rosa et al., 2014). BRCs have also gained attention from astrobiologists, who point out that such communities, when located in deserts, may act as analogues for possible life on Mars (Corenblit et al., 2019). Yet, despite the suggested a link between rock surface morphologies and microbial activity, one basic but important question remains unanswered: How quickly do BRCs develop after a rock surface has been exposed? Illustrating the origin, composition and growth rates of BRCs may shed light on the timescales where natural processes (e.g., desertification, land degradation) take place in arid regions. Currently, there is no available information about the development rates of epi-and endolithic lithic biofilms on rock surfaces in deserts. Formation rates of sub-aerial BRCs were noted to be very slow (Pointing and Belnap, 2012) but were not backed by numbers. The goal of this study was to provide a first quantitative estimation of the growth rate of lithic communities on rock surfaces in arid regions under natural conditions. Conventional dating methods like radiocarbon cannot be applied to these rock features due to the lack of datable material. Cosmogenic radionuclide methods may be useful by yielding maximum-limiting surface exposure estimates. However, they require large amounts of sample material, which is destructive and, therefore, unsuitable when dating archaeological sites or monuments. Lichenometry, which has been used extensively as a chronological tool in Arctic settings, is inapplicable for studying hot-desert geomorphology since deserts lack the moisture needed for lichen development (Dorn, 2009; Mcllroy de la Rosa et al., 2014). To overcome these obstacles, we used a well-preserved archaeological site built of stones that exhibit developed BRCs, marking the upper limit of the time frame needed for such processes to occur. The site was constructed from chalk and limestones, the dominant rock types in the Negev Desert, and in deserts in general (Mabbutt, 1977). Furthermore, because the nearby hillslopes were the source for the building stones, the features and composition of the BRCs can be compared to those under natural conditions. Owing to local environmental parameters, we hypothesised that both the BRC origin, composition and growth rates would differ between stones composing the archaeological buildings and natural stones in the adjacent slopes. To test our hypothesis, we combined field observations, geological, archaeological, and molecular microbiology characterisation to estimate the development rate and compositional trajectory of BRCs in arid regions.

## 1.1 Materials and methods

To estimate the BRC development rate, we applied a quantitative analysis comparing microbial communities from chalk and rocky limestone slopes to their equivalents in a dated archaeological site, the Byzantine city of Shivta (Fig. 1). Evaluating the origin and composition of the BRCs, the study involves archaeological, geological and microbiological methods on collected BRC coated rocks.

### 1.1.1 Study site

Shivta is an exceptionally well-preserved village of the Byzantine period (4th–7th centuries CE) that continued to be partially occupied into the Early Islamic period (8th-9th centuries CE; Avni, 2014). Shivta is located in the south-western regions of the Negev Desert, Israel, at an elevation of 350 m.a.s.l (30.88°N; 34.63°E; WGS 84 Grid; Fig. 1) where the environmental settings maintain arid conditions since the Holocene (10,000 years ago) and characterised by an average annual precipitation of 90 mm yr$^{-1}$ and an aridity index (P/PET) of 0.05, on the borderline between arid and hyperarid ecosystems (Amit et al., 2011). The site and its vicinity are part of a rocky terrain, underlined predominantly by carbonate rock slopes. Limestone bedrock outcrops of the Turonian Nezer Formation and soft chalk outcrops of the Santonian Menuha Formations surround the city (Fig. 1). BRCs are common on the atmospherically-exposed parts of many of the rocks. The site covers an area of 0.8 km$^2$ and contains the remains of three churches, a central watchtower and associated structures, two sizeable public water reservoirs, three wine-presses, a large inn of the Byzantine period and numerous private buildings. The main occupation of the residents in the Byzantine period appears to have been agricultural, mainly viticulture for the export of wine, and road services for Christian pilgrims travelling to and from Mt. Sinai (in modern-day Egypt). Thanks to these activities, the village appears to have been very prosperous in the Early and Middle Byzantine periods (4th-mid-6th centuries CE) when most of the village was constructed. Excavations in the site in 2016 conducted by a team led by G. Bar Oz and Y. Tepper on behalf of Haifa University has confirmed the date of the construction of private houses in the southeastern part of the town (Tepper et al., 2018). Their excavations extended to the bedrock under several houses over which the structures were constructed. The location of the village in an arid setting in the central Negev Highlands forced the inhabitants to construct buildings using locally quarried stone with minimum use of wood, which would otherwise have had to be imported from great distances. Therefore, the upper floors were supported with stone arches and wood was sparsely used for doorframes, doors and shelves. In lieu of wood, many installations, such as animal troughs, were carved from local stone. The walls of the houses are well-preserved, offering researchers a unique view of buildings constructed over 1700 years ago that stand to heights of two and even three stories high. The lower courses of the walls on the ground floor were made using rough limestone blocks taken from the Nezer Formation, while the upper walls were constructed of a lighter chalkstone, taken from Menuha Formation that is more easily worked into blocks. The spaces between the heavy stones of the lower courses were sealed with mortar, and the interior walls were often covered with a base of mud plaster covered with white lime plaster (Fig. 2).

**Table 1.** Biological rock crust (BRC) thickness and geotechnical properties of the subjected strata (mean ± SE

| Rock properties | | Chalk[1] | Limestone[2] |
|---|---|---|---|
| BRC mineralogy (%) | Calcium | 92 | 95 |
| | Quartz | 2 | 4 |
| Host rock mineralogy (%) | Calcium | 95 | 95 |
| | Quartz | 0 | 0 |
| Host rock porosity (%) | | 26 | 7.1 ± 0.7 |
| Host rock bulk density (g cm-3) | | 1.95 ± 0.1 | 2.5 ± 0.1 |
| BRC thickness ($\mu$m) n=6 | Shivta Byzantine city | 290 ± 74 | 3206 ± 258 |
| | Bedrock slopes | 1279 ± 175 | 2585 ± 380 |

[1] The chalk was dated to the Menuha formation

[2] The limestone was dated to the Nezer formation

### 1.1.2 Field sampling

A total of 24 rock samples were collected from Shivta Byzantine city (30.88°N; 34.63°E; WGS 84 Grid; the samples were named ShivtaSite 1-12) and its surroundings (30.87°N; 34.62°E; WGS 84 Grid; the samples were named ShivtaSlope 1-12). Twelve rock samples were collected from the limey Nezer Formation including six samples from the archaeological site and six from the nearby natural slope. The same sampling procedure was applied for the chalky Menuha Formation. All rock samples taken from the slopes and site walls included whole rocks collected directly in the field and were not subsampled in the lab. To avoid the slope aspect effect that may lead to different moisture regime, all samples were retrieved from south-facing slopes/walls and were collected during January 2015.

### 1.1.3 Geological analysis

The geological methods used in this study were applied to estimate the potential role of the substrate. The methods based on direct field observations and characterisation of the subjected lithologies using thin sections, XRD analyses, total effective porosity and stable isotope analysis. Petrographic thin sections, 30 $\mu$m thick, were prepared for each lithology to test the main components in both the rock crust and the host rocks examined under a light microscope (Zeiss, Oberkochen, Germany). XRD analyses for bulk mineralogical components (Sandler et al., 2015) were conducted separately on the rock crust and the host rock. Three replicates were collected from each lithology. Powdered samples were scanned by a PANALYTICAL X'Pert3 Powder diffractometer equipped with a PIXcel detector. The scanning range was: 3–70° $2\theta$, step size 0.013°, speed 70.1 s per step. Total effective porosity ($\phi$) (Scherer, 1999) was performed using a gas permeameter and porosimeter device (Core laboratories, Amsterdam, Netherlands) on 12 rock core cylinder samples (radius 18.5 mm and height 26.5 mm). The value of chalk total effective porosity was cited from Schütz et al., (2012) and reported in Table 1.

For the stable isotopes $\delta^{13}C$ and $\delta^{18}O$ analyses, 1-2 mg of rock surface powder was obtained using a microdrill (Dremel, Racine, WI, USA) along a cross section of the rock crust and its host rock. Six profile measurements of $\delta^{13}C$ and $\delta^{18}O$ were performed on the chalk and limestone samples. Measurements (in duplicate) of $\delta^{18}O$-$H_2O$ and $\delta^{13}C$-DIC were performed on a gas source isotope ratio mass spectrometer (GS-IRMS; Thermo Fisher Scientific, Waltham, MA, USA) coupled to a Gas Bench II interface (Thermo) after $CO_2$ equilibration or $CO_2$ extraction by acidification for $\delta^{18}O$-$H_2O$ and $\delta^{13}C$-DIC, respectively. The samples were calibrated against internal laboratory standards: Vienna Standard Mean Ocean Water (VSMOW) and carbonate standard NBS19. $\delta^{13}C$ and $\delta^{18}O$ values were also referenced relative to Vienna PeeDee Belemnite (VPDB) standard as previously described (Uemura et al., 2016) with SD of 0.1‰. All values are reported in per-mil (‰).

### 1.1.4 Method validation

Method validation was conducted at Nitzana archaeological site using the same procedure as for the Shivta site. The ancient village of Nitzana (Nessana), ca. 270 m asl, is situated in the western part of the Negev Desert (Fig. S1), ca. 20 km from the Shivta site. Excavation at the site revealed that it was first established during the Hellenistic period and continued to be occupied through the Roman, Byzantine and Early Islamic periods (Langgut et al., 2020 and reference therein). The excavation at the Nitzana site exposed domestic quarters, two churches, a fortress and an archive of papyri from the 6–7th centuries CE. The papyri make Nitzana the best-documented community of all Byzantine-Early Islamic sites in the Negev. The lower courses of the walls at the site were made using limestone blocks taken from the Lower Yeter Formation, while the upper walls were constructed of a lighter chalkstone, taken from the Upper Yeter Formation (Fig. S1).

### 1.1.5 DNA extraction PCR amplification and sequencing

For DNA extraction from rocks, the surface (ca. 100 cm$^2$) was scraped off using a wood rasp (66-67 HRC hardness; Dieter Schmid, Berlin, Germany) that was cleaned with 70% technical-grade ethanol before each sampling. DNA was then extracted using 0.4 g of a sample using Exgene Soil DNA extraction kit (GeneAll, Seoul, S. Korea) according to the manufacturer's instructions. A 466-bp fragment of the SSU rRNA gene was amplified using the universal bacterial primers 341F (CCTAYGGGRBGCASCAG) and 806R (GGACTACNNGGGTATCTAAT) flanking the V3 and V4 region (Klindworth et al., 2013). Library construction and sequencing were performed at the DNA Services Facility, the University of Illinois at Chicago (USA) using a MiSeq sequencer (Illumina, San Diego, CA, USA) in the $2 \times 250$ cycle configuration (V2 reagent kit). The raw SRA files were deposited into EMBL- ENA SRA database (https://www.ebi.ac.uk/ena/) and can be found under study accession PRJNA381355.

### 1.1.6 Sequence processing and analysis of bacterial communities

Paired reads generated by the MiSeq platform were quality filtered and clustered into OTUs using the UPARSE pipeline (Edgar, 2013), with modifications. Contig assembly was done using the fastq_mergepairs command. Then, contigs were dereplicated

with the derep_fullength command, and singleton sequences were removed. OTU centroids were then determined with the cluster_otus command (set at 3% radius). Abundances of OTUs were determined by mapping the filtered contigs (before dereplication, including singletons) to the OTU centroids using the usearch_global command (set at 0.97% identity). Following these steps, a total of ca. 1.4 million reads remained. OTU representatives were classified using mothur's implementation of a Naïve Bayesian sequence classifier (Schloss et al., 2009; Wang et al., 2007) against the SILVA 119 SSU NR99 database (Quast et al., 2013). All downstream analyses were performed in R V3.4.4 (R Core Team, 2020). Data handling and manipulation were done using package phyloseq (McMurdie and Holmes, 2013). For alpha-diversity analysis, all samples were subsampled (rarefied) to the minimum sample size using a bootstrap subsampling with 1000 iterations to account for library size differences, while for beta-diversity analysis library size normalisation was done using GMPR (Chen et al., 2018). The Inverse Simpson's, Shannon's H diversity indices, and the Berger-Parker dominance index were calculated using the function EstimateR in the vegan package (Oksanen et al., 2018) and tested using ANOVA in the stats package, followed by a *post hoc* Estimated Marginal Means test (Searle et al., 1980) from the emmeans package (Lenth et al., 2020). Variance partitioning and testing were done using PERMANOVA (McArdle and Anderson, 2001) function vegan::adonis using Horn-Morisita distances (Horn, 1966). Differences in phyla composition between the sample-type were tested using the non-parametric Aligned Rank Transformation ANOVA (ART ANOVA; Wobbrock et al., 2011) in the package ARTool and FDR corrected using the Benjamini-Hochberg method (Ferreira and Zwinderman, 2006) (function stats::p.adjust). Detection of differentially abundant OTUs was done using a beta-binomial regression model (Martin et al., 2019) in the package corncob. Plots were generated using packages ggplot2 (Wickham, 2016). The scripts for reproducing the microbial analysis can be found at: https://github.com/roey-angel/BRC_growth_microbiome.

## 1.2 Results

### 1.2.1 Geotechnical properties

BRCs were observed on all rock surfaces (i.e., limestone and chalk) both in the Byzantine site of Shivta and adjacent slopes. To evaluate the geological differences between BRCs, we performed geological characterisation of 24 limestone- (Nezer Formation) and chalk- (Menuha Formation) samples. Table 1 depicts the geotechnical parameters tested, including mineralogy, total effective porosity and bulk density. Several large quarries correlated with the Byzantine period (i.e., the Byzantine rule over Syria-Palestine 390-636 AD) are found within the chalk and limestone slopes of the Menuha and Nezer Formations in the northern Negev Desert (Fig. 1).

The limestone rocks constructing the Byzantine site were carved from the Turonian Nezer Formation; a well-bedded (30-50 cm) bio-micritic, fine to coarse-crystalline. Being well-bedded, limestone is easily useable as a building material for different purposes (walls and ceilings). These blocks were only lightly polished since they were used mainly as the foundation of walls, leaving their original BRC intact. Consequently, most rocks were irregular in shape and sealed with mortar covered with white lime plaster (Fig. 2).

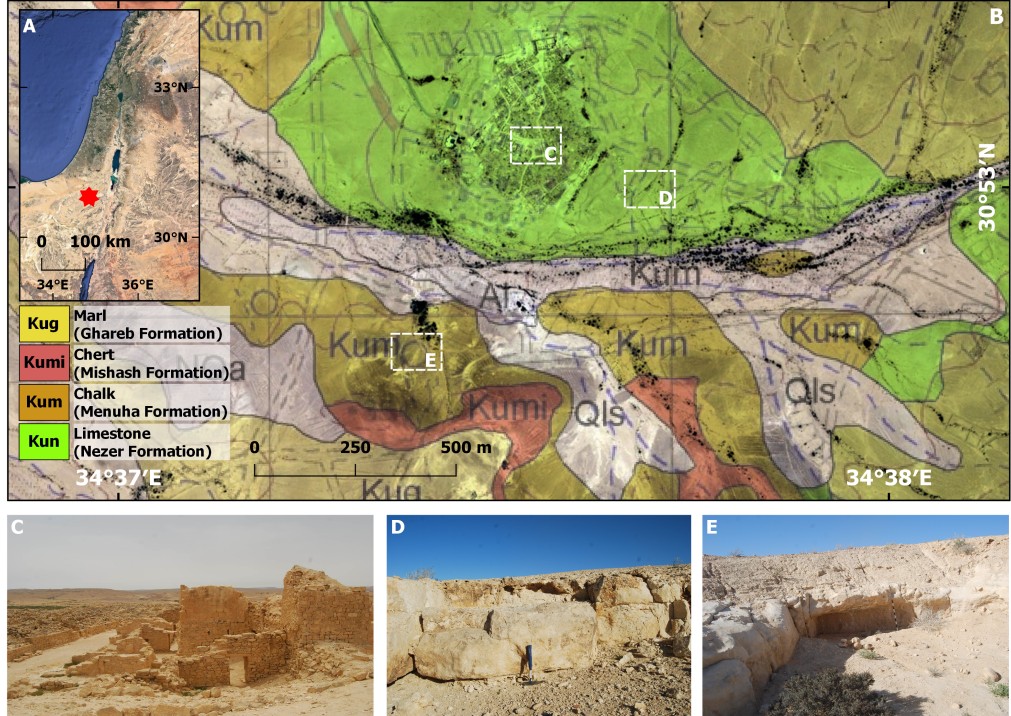

**Figure 1.** (A) Geographical location of the study area (red star) in the central Negev Desert, Israel (Google, ©2020 Landsat / Copernicus Data SIO, NOAA, U.S. Navy, NGA, GEBCO Mapa GISrael); (B) Shivta geological map (Sneh et al. 2011; reproduced with permission) (C) A panoramic view of Shivta Byzantine city; (D) Ancient limestone quarry located at the vicinity of Shivta Byzantine city (30 cm hammer for scale); (E) Ancient chalk quarry located at the vicinity of Shivta Byzantine city (1 m black and white stick for scale).

The chalk blocks were carved from the Santonian Menuha Formation: a soft white massive chalk found applicable as a building material thanks to the ease of polishing it. New rock surfaces were exposed in the carving process while building Shivta, leaving no remnants of the original BRCs Soft chalk blocks were used mainly for building the upper parts of the walls (Fig. 2). The absence of plaster between the chalk blocks, unlike the limestone blocks, indicates the intense polishing of these building blocks in order to fit them together. Therefore, the carved chalk blocks could be considered succession planes for new microbial colonisers.

### 1.2.2 Thickness measurements

Microscopic examinations of thin sections (30 $\mu$m thick) prepared from 24 limestone and chalk blocks showed that, as expected, the BRCs were restricted to the atmospherically-exposed parts of the rocks (Wieler et al., 2019). The BRCs were characterised by a hardpan-laminated structure composed of masses of micritic to microsparitic carbonate layers interbedded with microbial

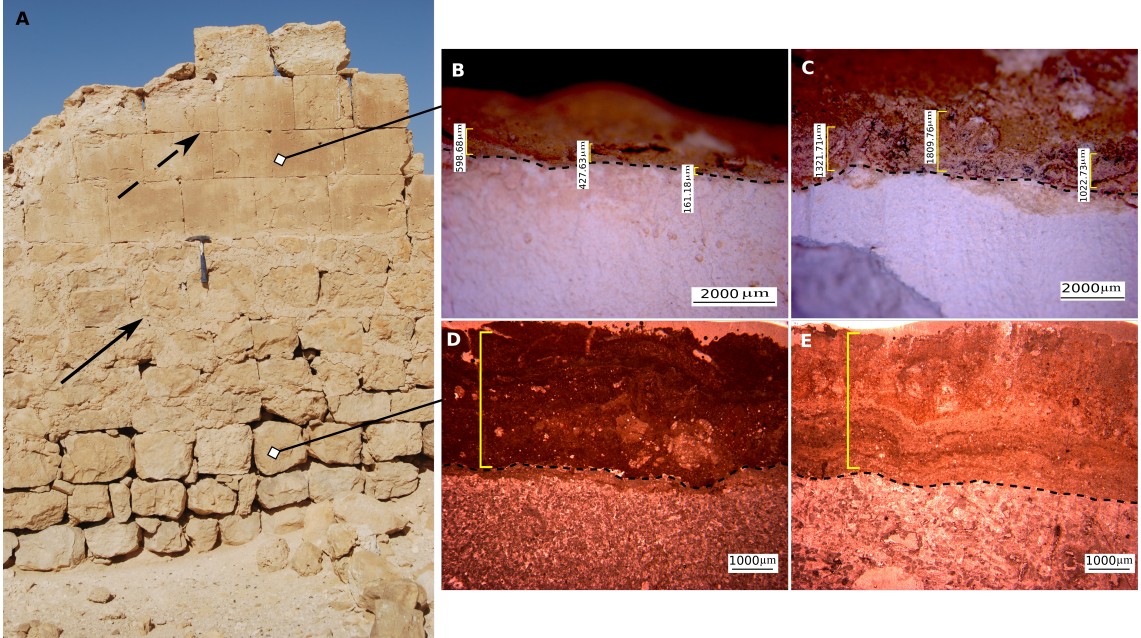

**Figure 2.** (A) Two rock types were retrieved from a south-facing exterior wall (30 cm hammer for scale) in Shivta Byzantine city (1700 years old). The upper section of the wall was constructed of chalk blocks (dashed arrow), while the lower section was constructed of limestone blocks joined by plaster (solid arrow); (B) A thin BRC as observed in the chalk blocks from Shivta Byzantine city compared to a thicker BRC as observed in an uncarved chalk rock from the adjacent slopes(C); (D, E) Limestone BRCs as observed in a sample collected from the city (D) and adjacent slopes (E). Dashed lines mark the border between the BRC at the top and the host rock at the bottom.

coatings covering the lime and chalk host rocks. BRC thickness significantly differed between those found on limestone compared to chalk building blocks, even when located on a single wall in the archaeological site (Table 1, Fig. 2). Limestone BRC thickness at both the Byzantine site and along the rock outcrops located in the limestone quarries at the adjacent slopes ranged between 1.4–4.3 mm ($3.2 \pm 0.26$ and $2.6 \pm 0.38$; Fig. 2D, 2E). In contrast, chalk BRC thickness differed between the blocks in the Byzantine site and the natural chalk slopes. The chalk BRC thickness at the Byzantine site ranged between 0.1–0.6 mm ($0.29 \pm 0.07$; Fig. 2B), while at the adjacent slopes, it ranged between 0.5–1.8 mm ($1.28 \pm 0.18$ Fig. 2C). Dating Shivta site is mainly associated with the early-middle Byzantine period (4$^{th}$–6$^{th}$ centuries CE)(1700-1500 years ago) as was recently confirmed by G. Bar Oz and Y. Tepper on behalf of Haifa University (Tepper et al., 2018). Dividing the observed thickness of the BRC from the chalk building blocks by the age of Shivta Byzantine site (1700-1500 years), located under long stable arid conditions and free of anthropogenic effects, suggest a growth rate of 0.06–0.35 mm 1000 yr$^{-1}$ ($0.17 \pm 0.04$).

## 1.3 Method validation

We validated our BRC dating method by conducting a parallel thickness measurement for BRCs sampled from a different archaeological site at Nizzana (see Materials and Methods). We used an identical procedure as in the Shivta site. All samples

were retrieved from south-facing walls and were processed similarly. We measured rock crust thickness of 0.1-0.2 mm (1.67 $\pm$

195  0.01) on the carved limestone and chalk building blocks. Dividing the thickness by the age of the Nitzana site, we found the

BRCs growth rates consistent with the one calculated at the Shivta site (0.1 mm 1000 yr$^{-1}$).

### 1.3.1  Isotopic composition

The biogenic nature of the crusts was confirmed using a cross-section analysis of the stable carbon and oxygen isotope ratios

in the crust and host rock. For the limestone sample, a shift was found between the $\delta^{13}$C values for the BRC (0–2 mm) and

200  the host rock (2–5 mm) layers, with BRC values ranging between -4‰ and -5‰ VPDB and host-rock values between 0‰ and

1‰ VPDB(Fig. 3A). However, for the chalk, $\delta^{13}$C values for the BRC ranged between -0.2‰ and -1.9 ‰ VPDB, and between

0.1‰ and–2.7‰ VPDB in the host rock. These results were consistent for both the slope and archaeological site samples. The

limestone $\delta^{13}$C values are typical indicators of carbon isotope exchange of primary marine $CaCO_3$ (abundant in the bedrock)

with $CO_2$ released by microbial respiration (i.e. of carbon originating from photosynthesis) with the subsequent precipitation

205  of pedogenic calcrete (Brlek and Glumac, 2014). The differences in $\delta^{13}$C values between the chalk and limestone are suggested

to result from the BRC thickness. Thicker BRCs, as observed in the limestone samples, should produce more biogenic activity

than the chalk. The limestone $\delta^{18}$O values ranged between -3‰ in the BRC to -7.3‰ VPDB in the host rock. This decrease

in $\delta^{18}$O values in the host rock is suggested to result from meteoric water substitution of the marine limestone (Alonso-Zarza

and Tanner, 2006). The chalk $\delta^{18}$O values ranged between -3.3‰ and -5.4‰ VPDB for both the BRC and the host rock, these

210  negative $\delta^{18}$O values were consistent with other Santonian chalk (Clarke and Jenkyns, 1999; Liu, 2009).

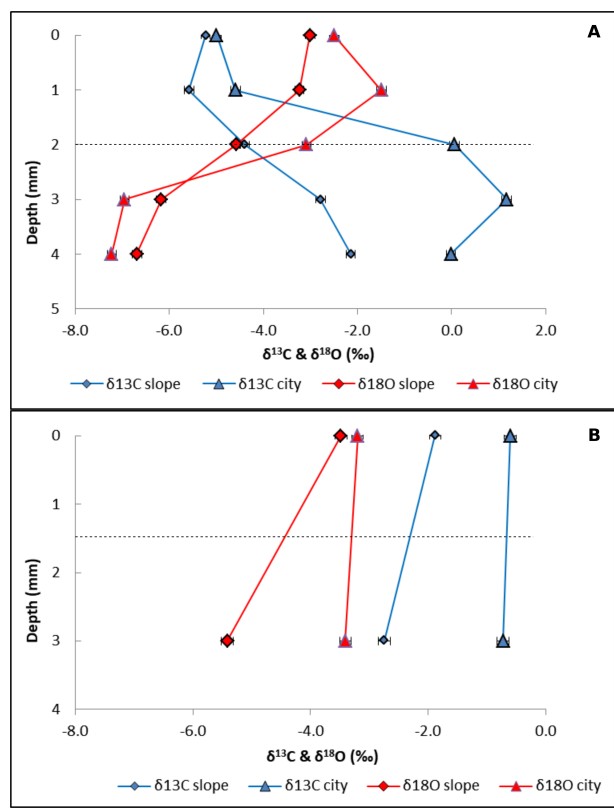

**Figure 3.** Carbon and oxygen isotopes profiles in BRCs and host rock samples of limestone (A) (Nezer Formation) and chalk (B) (Menuha Formation) samples collected from the Shivta site (triangles) and adjacent slopes (diamond). Dashed lines mark the border between the BRC at the top and the host rock at the bottom.

### 1.3.2 Bacterial diversity of the BRCs in the Shivta Byzantine site and the nearby slopes

To elucidate the identity of the bacterial communities on the BRCs, we performed multiplexed barcoded amplicon sequencing of the small subunit ribosomal RNA gene (SSU rRNA gene). As expected, we found simple BRC communities (i.e., low-richness and low-diversity) in all BRC samples. Comparing the BRCs collected from the Byzantine site to the nearby slopes from the two primary lithologies (the limestone and chalk) showed no statistically significant differences in the observed number of OTUs or their inverse Simpson or Shannon indices, with all samples averaging $312 \pm 11$ OTUs (Fig 4A, Table S1). However, the city samples showed a slightly higher dominance of the most abundant OTU compared to the slope samples (BP values of $0.24 \pm 0.03$ vs $0.14 \pm 0.01$; p=0.0017). The bacterial community composition of the BRCs was very similar in both lithologies and sample sources and was heavily dominated by members of the phylum Actinobacteria, followed by Cyanobacteria, Proteobacteria (mainly Alphaproteobacteria from the orders Sphingomonadales and Rhodospirillales) and Bacteroidetes (Fig. 4C). A variance partitioning analysis did, however, show a small difference between the samples from the city and the ones from the slopes, explaining 8.1% of the variance in the Morisita-Horn distance matrix (p=0.023). In contrast, neither the lithology nor the interaction between the lithology and sample source correlated with differences in bacterial community composition (Table S2). The difference in the communities between the samples from the city and the ones from the slopes were not detected when comparing the dominance of each phylum between locations using Aligned Rank Transformed ANOVA test (Fig 4D). However, using a differential abundance test, 64 individual OTUs were differentially more abundant in the city samples while 57 were differentially more abundant in the slope samples (from a total of 694 OTUs; Fig. S2). These OTUs came from all the dominant phyla and belonged to orders that are commonly found in hot deserts such as Rubrobacterales and Solirubrobacterales (Actinobacteria), Cytophagales (Baceroidetes), Thermomicrobiales (Chloroflexi), Chroococcidiopsidales (Cyanobacteria), Rhodospirillales (Alphaproteobacteria), and Burkholderiales (Betaproteobacteria) (Table S3, Angel and Conrad, 2013; Makhalanyane et al., 2015). However, there does not seem to be a discernible taxonomic pattern for the differentially abundant OTUs.

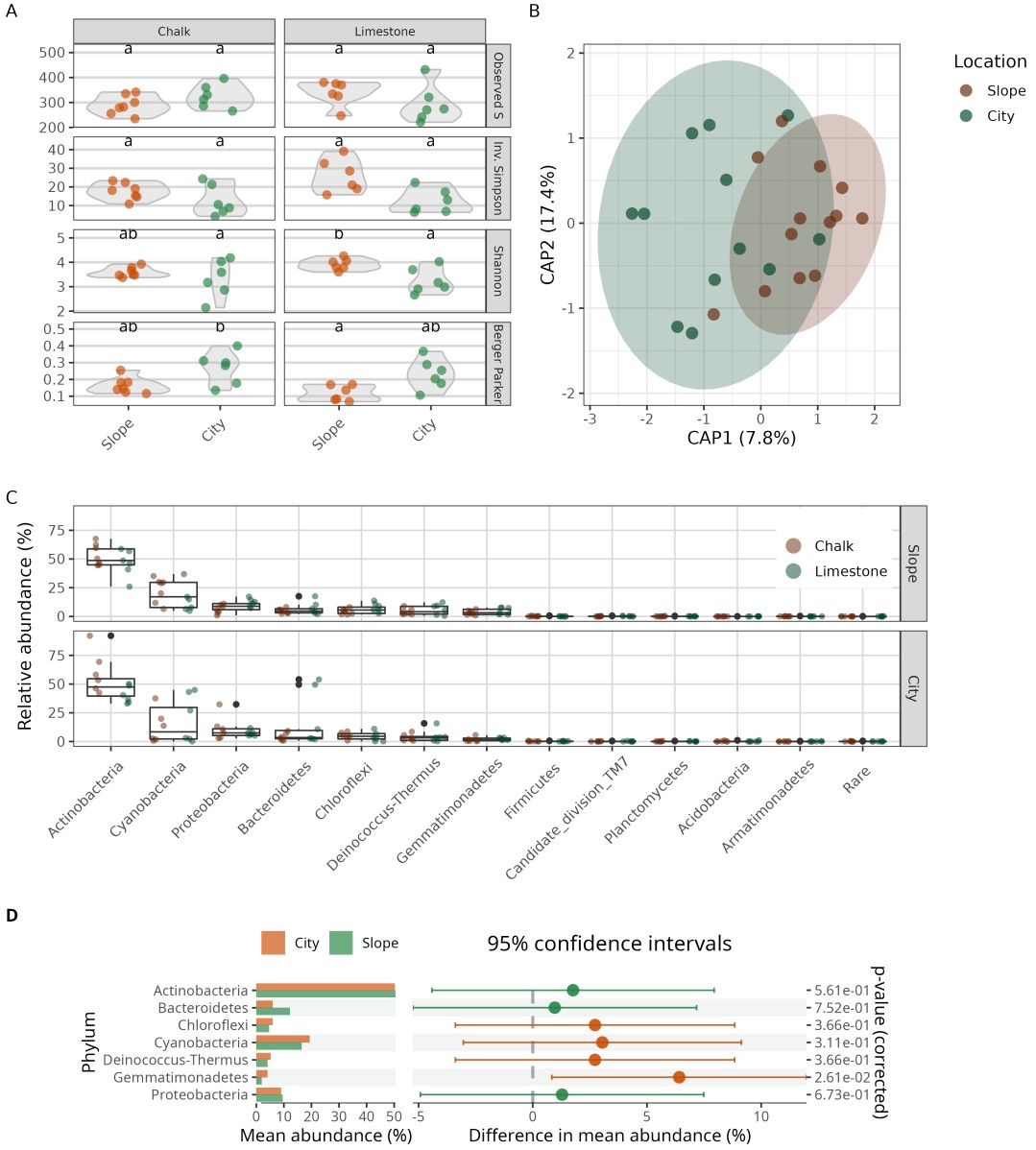

**Figure 4.** (A) Bacterial alpha-diversity indices (Observed S, Inv Simpson, Shannon and Berger Parker) for chalk and limestone BRCs. Different lower-case letters denote statistical difference at $P < 0.05$. (B) Bacterial community composition in the chalk and limestone BRCs sampled from the Shivta Byzantine city and adjacent slopes. 'Rare' denotes all OTUs belonging to phyla that account for less than 5% of the combined relative abundance. (C) Canonical analysis of principal (CAP; i.e. constrained principal coordinates analysis) method showing the differences in bacterial community composition in limestone and chalk BRCs collected from the city and nearby slopes (n=24). The numbers in brackets indicate the explained variance in the constrained model (Morisita-Horn distance as a function of location). (D) Differences in mean abundance, 95% confidence intervals and statistical significance (based on ART-ANOVA) on the phylum level between slope and city samples (only phyla that account for more than 5% of the relative abundance are shown).

## 1.4 Discussion

BRCs are a common and important feature of atmospherically-exposed rock surfaces in drylands around the globe, yet reliable growth estimations based on field data are rare. Only a handful of studies tried to estimate BRC growth rates, among them Lange (1990) measured radial growth rates of epilithic lichens in the arid Negev Desert, Israel, and reported an average growth rate of 0.371 mm yr$^{-1}$. Another study conducted by Krumbein and Jens (1981) found that desert varnish, a typical rock crust in desert regions, show black fungi microcolonies after eight weeks of isolation in the lab. Liu and Broecker (2000) noted the accumulation rates of desert varnish in natural conditions to be much slower, in the range of <0.001 to 0.04 mm kyr$^{-1}$, and rarely reach thickness exceeding 200 $\mu$m.

In this study, we observed a 0.1 to 0.6 mm thick biological rock crust coverage on chalk building blocks of the Byzantine site (Fig. 2B) dated to the 4$^{th}$–7$^{th}$ centuries CE (Tepper et al., 2018). The building blocks experienced long stable arid conditions and were free from direct anthropogenic influences. Thanks to the processing method of the bricks used for construction, leaving BRC-free surfaces exposed to the elements, the BRC is confined by the dating of the archaeological site (Tepper et al., 2018). This provides us with an estimated growth rate of 0.06–0.35 mm 1000 yr$^{-1}$ for BRCs of primarily bacterial origin. Besides, the presence of a 1.4–4.3 mm thick BRC in all limestone rocks (Fig. 2D, 2E) and 0.5–1.8 mm on the chalk rocks collected from the slopes (Fig. 2C) provide a plausible maximum for BRC growth under these conditions. Calibrating local climate conditions and mineral composition is needed for BRCs growth rates when applying to different conditions. The presence of similarly thick crusts on the rocks of a different site which is dated to the same period but is located 20 km away and is constructed of slightly different rocks, stemming from a different formation, provides validation for our methods.

Applying stable isotopes, we noted a consistent trend where more negative $\delta^{13}$C values were found in the limestone crust compared to their host rock (Fig. 3A), reflecting a biogenic agency in the rock crust production, regardless of the substrate type. The oxygen isotopes profile of the chalk samples collected from the archaeological site showed no difference between the bedrock and the crust (Fig. 3B) Thus, it indicates the early stages of BRC formation. The limestone samples showed more negative $\delta^{13}$C values in the BRC compared to the host rock. Yet, the $\delta^{18}$O values were more negative at the host rock compared to the BRC, which may indicate sub-aerial exposure of the BRC and substitution of the marine limestone.

From a microbiological perspective, all the samples studied here showed very similar microbial community composition, irrespective of lithology or BRC thickness. This similarity in composition demonstrates the indifference of microorganisms to the type of attachment surface in this case and that the community probably changes very little after establishing. However, the differences in BRC thickness between the chalk and limestone sampled from the slopes could indicate that the latter can better support BRC growth.

Differences in the dominance of the most abundant OTU as well as some differences in the relative abundance of about 17% of the OTUs were observed between BRCs from the city and slope samples. The OTUs belonged to taxa that are commonly found in hyperarid deserts. However, the differences in abundance were relatively minor. Moreover, they lacked a clear taxonomic pattern, with different OTUs belonging to the same taxa ending up as either differentially more or less abundant. This could indicate a strong influence of stochastic processes (founder effect) over environmental filtering at the OTU level. Therefore,

our findings indicate that the rock surfaces at the archaeological site act as succession planes for BRC development. Moreover, similarities in the microbial composition between rocks that were in contact with the ground to those that were detached from the ground (chalk BRCs found in Shivta) indicate a major role to aeolian processes (e.g., settled dust particles) in determining the community composition of BRCs in deserts as was previously reported (Wieler et al., 2019).

Assisted recovery of lithobiontic communities has not been conducted in natural settings, and most research focused on the regeneration of soil biocrust (Velasco Ayuso et al., 2017). Artificial cultivation of soil biocrusts (Zhang et al., 2018) suggested that inoculation of cyanobacteria and algal communities enhance the recovery of biocrusts. Testing the soiling impact on rock surfaces indicated that bacterial colonisation plays an important part in the development of fungal biofilms (Viles and Gorbushina, 2003). In fact, cyanobacteria and fungi are believed to be critical components of soil and rock biocrusts (Gorbushina, 2007; Weber et al., 2016). In arid BRCs, fungi are scarce (Lang-Yona et al., 2018), but our results suggest that not only Cyanobacteria but other taxa like Alphaproteobacteria and Actinobacteria play a key role in BRC development (Fig. 4C). Similarly, the dominance of Alphaproteobacteria and Actinobacteria in early soil biocrusts formation was observed in arid regions (Ji et al., 2017; Pepe-Ranney et al., 2016).

Unlike more humid dryland soils, in hyperarid soils and rock biocrusts, Actinobacteria remain the dominant phylum even in later stages of development (Holmes et al., 2000; Angel and Conrad, 2013; Idris et al., 2017). Kuske et al. (2012) identified a deep branching subclass within the Actinobacteria phylum abundant worldwide, the Rubrobacteridae, a taxon extremely resistant to desiccation and UV stress (Holmes et al., 2000; Rainey et al., 2005). The presence of Rubrobacteridae suggests that they may be involved in shaping the rock surface structure during biocrust formation (Mummey et al., 2006; Rainey et al., 2005).

Our analyses suggest that both rocks experience the same regional-scale environmental factors (Fig. 4B)in shaping the BRC composition. This demonstrates the ecological filtering effect of the rock surfaces, which imposes unique abiotic challenges for the inhabiting microbes, and infer that dust particles are the main potential source of colonisation the microbial communities (Wieler et al., 2019). We thus suggest that local-scale environmental parameters play a major role in shaping the microbial taxa that colonise fresh rock surfaces in arid regions. This also suggests that the BRCs cannot be regarded as passive deposits of microbial cells but should rather be seen as a filter selecting for a specific subset of adapted microbes that can persist and form a biofilm under these harsh conditions.

## 1.5 Conclusions

This study provides the first estimate to an unfathomed question on the rate of BRC growth rate under a natural setting. The growth rate observed here, 0.06-0.35 mm 1000 $yr^{-1}$, validates the extremely slow nature of such a succession process. Using a well-dated archaeological site in the arid Negev Desert, Israel, we demonstrate the possibility of using such human-made artefacts to document and confine long microbial developmental processes that are otherwise too slow to monitor. We confirm the biological origin of the crusts through analysis of stable carbon and oxygen isotope ratios. We further show that local-scale environmental parameters are found to shape the microbial taxa that colonise the fresh rock surfaces, including Actinobacteria, Cyanobacteria and Proteobacteria. It also suggests that the BRCs cannot be regarded as passive deposits of microbial cells but

should rather be seen as a filter selecting for a specific subset of adapted microbes that can persist and form a biofilm under these harsh conditions. Conversely, once a growth rate has been established for a region, it could be used, by itself, to date the age of atmospherically exposed archaeological artefacts.

*Code availability.* https://github.com/roey-angel/BRC_growth_microbiome

305 *Data availability.* https://www.ncbi.nlm.nih.gov/bioproject/PRJNA381355

# Appendix A: Supplementary figures and tables

**Table A1.** ANOVA test results for various alpha-diversity metrics.

| Observed S | Sum Sq | Df | F value | Pr(>F) | |
|---|---|---|---|---|---|
| (Intercept) | 588166 | 1 | 198.58 | 3.60E-12 | *** |
| Location | 3993 | 1 | 1.35 | 0.26 | |
| Rock.type | 7767 | 1 | 2.62 | 0.12 | |
| Location:Rock.type | 10123 | 1 | 3.42 | 0.08 | |
| Residuals | 62200 | 21 | | | |
| Inv. Simpson | | | | | |
| (Intercept) | 4281315 | 1 | 1306.61 | 2E-16> | *** |
| Location | 4 | 1 | 0 | 0.97 | |
| Rock.type | 1671 | 1 | 0.51 | 0.48 | |
| Location:Rock.type | 8756 | 1 | 2.67 | 0.12 | |
| Residuals | 68810 | 21 | | | |
| Shannon's H | | | | | |
| (Intercept) | 308.9 | 1 | 1353.08 | 2E-16> | *** |
| Location | 1.4 | 1 | 6.1 | 0.02 | |
| Rock.type | 0.1 | 1 | 0.36 | 0.55 | |
| Location:Rock.type | 0.3 | 1 | 1.15 | 0.3 | |
| Residuals | 4.8 | 21 | | | |
| Berger Parker | | | | | |
| (Intercept) | 7346 | 1 | 145.56 | 6.60E-11 | *** |
| Location | 543 | 1 | 10.76 | 3.60E-03 | ** |
| Rock.type | 99 | 1 | 1.96 | 0.18 | |
| Location:Rock.type | 117 | 1 | 2.32 | 0.14 | |
| Residuals | 1060 | 21 | | | |

**Table A2.** Variance partitioning of the Morisita-Horn distance matrix using PERMANOVA

| Observed S | Df | Sums of Sqs | Mean Sqs | F Model | R2 | Pr(>F) | |
|---|---|---|---|---|---|---|---|
| Location | 1 | 0.63 | 0.63 | 2.12 | 0.08 | 0.02 | * |
| Rock.type | 1 | 0.39 | 0.39 | 1.3 | 0.05 | 0.23 | |
| Location:Rock.type | 1 | 0.49 | 0.49 | 1.63 | 0.06 | 0.1 | |
| Residuals | 21 | 6.25 | 0.3 | 0.81 | | | |
| Total | 24 | 7.75 | 1 | | | | |

**Table S3.** Taxonomic classification and differential-abundance test results between the samples from the archaeological site (City) and the nearby slopes (Slope).

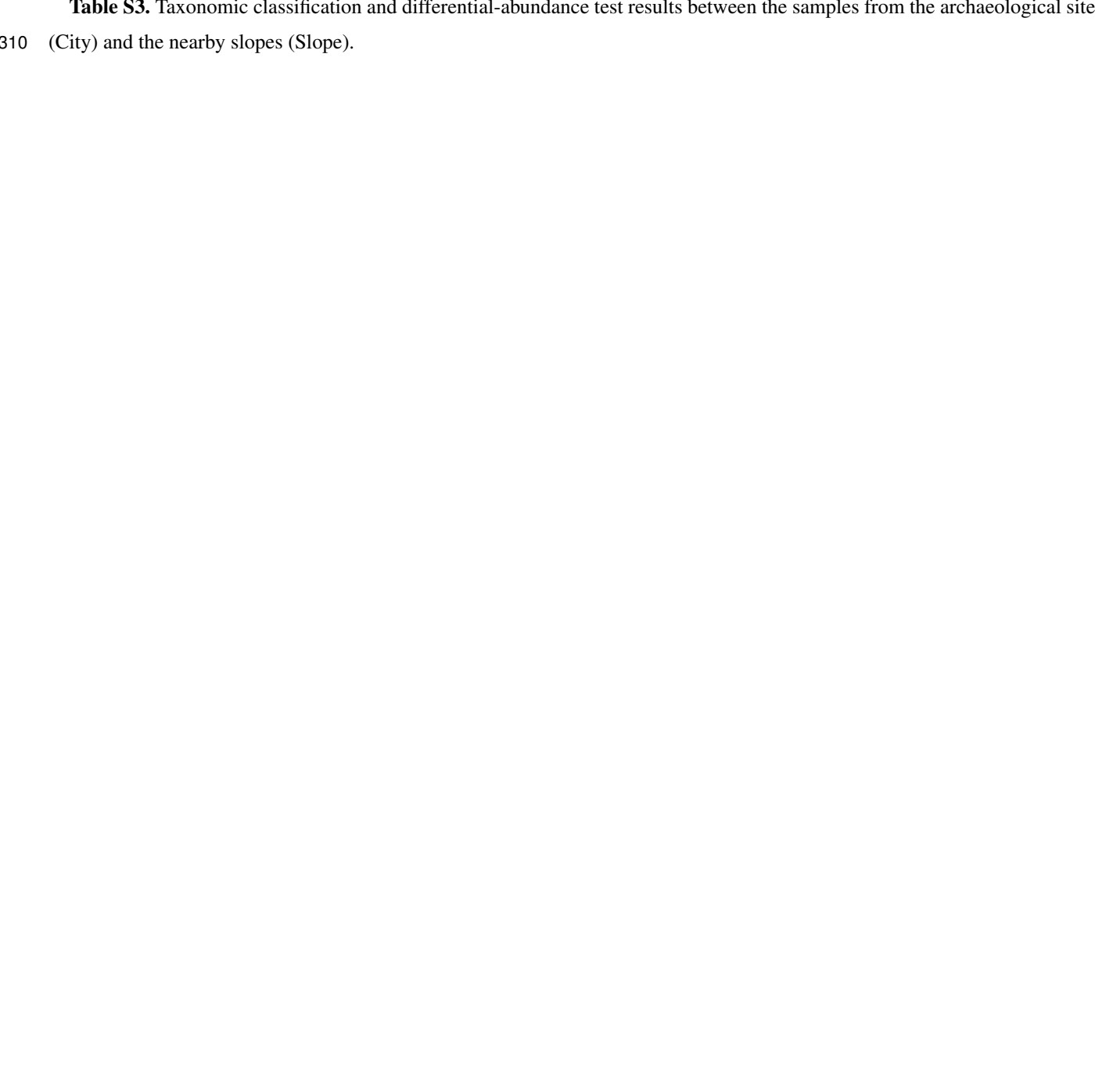

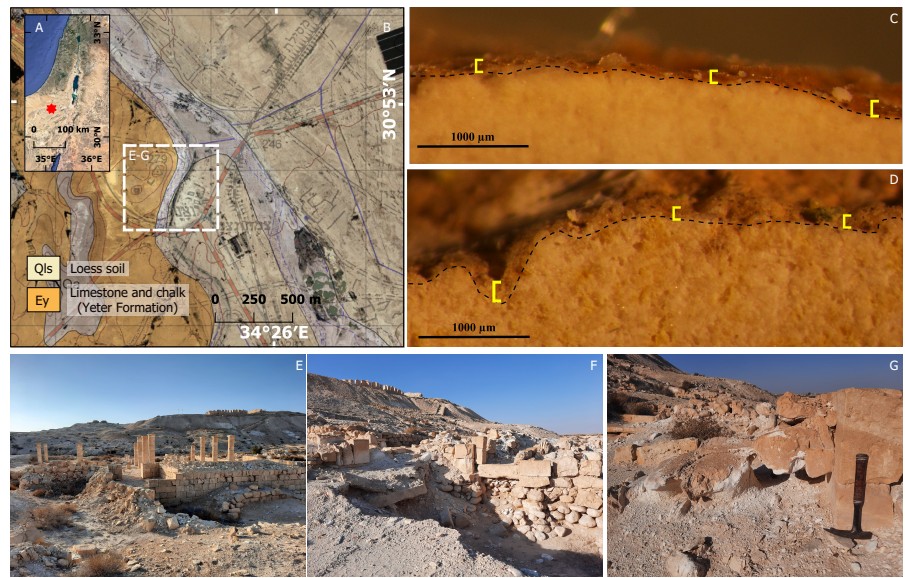

**Figure A1.** (A) Geographical location of the study area (red star) in the central Negev Desert, Israel (Google, ©2020 Landsat / Copernicus Data SIO, NOAA, U.S. Navy, NGA, GEBCO Mapa GISrael) ; (B) Nitzana geological map (Zilberman et al. 2011; reproduced with permission) ; (C) A panoramic view of Nitzana Byzantine city; (D, E) Buildings blocks at Nitzana site (30 cm hammer for scale); (F) BRC on chalk sample retrieved from the site, black dashed line marks the interface between BRC at the top and host rock at the bottom; (E) BRC on limestone sample retrieved from the site, black dashed line marks the interface between BRC at the top and host rock at the bottom.

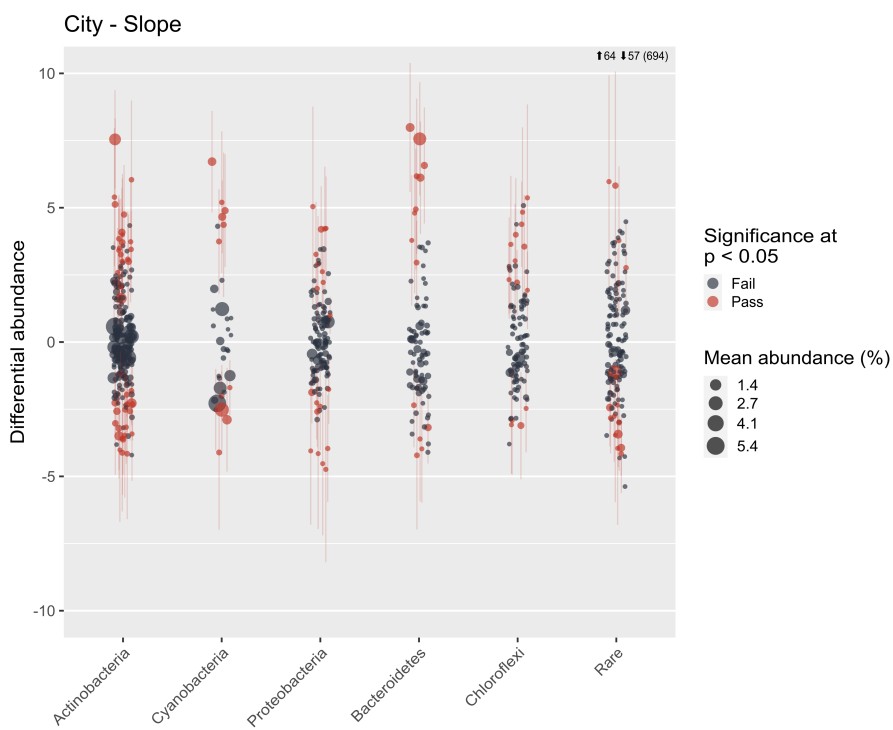

**Figure A2.** Detection of differentially abundant OTUs between the city and slope samples using a beta-binomial regression model (corncob). Each circle denotes a single OTU, and its size is its average relative abundance across all samples. The x-axis shows the classification of each OTU, whereas the y-axis denotes the difference in the modelled mean relative abundance between the city and slope samples. 'Rare' denotes all OTUs belonging to phyla that account for less than 5% of the relative abundance. Red circles are OTUs that show significant differential abundance at the P <0.05 level. Numbers next to the arrows (top right) indicate the number of significant differentially abundant OTUs that are either more abundant (up arrow) or less abundant (down arrow) in the city samples compared to the slope samples. The number in brackets indicates the total number of OTUs tested.

*Author contributions.* NW, TG, OG and RA conceptualised the study; NW performed the field and lab work; NW and RA analysed the data; NW, TG, OG and RA wrote the paper. OG and RA contributed equally to this study.

*Competing interests.* The authors declare no competing interests.

*Acknowledgements.* RA was supported by BC CAS, ISB & SoWa RI (MEYS; projects LM2015075, EF16_013/0001782–SoWa Ecosystems Research).

315

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
