# Peer review of "Microbial and geo-archaeological records reveal the growth rate, origin and composition of desert rock surface communities"

_Biogeosciences, 2020_

## Author Comment (AC1)

Dear Editor,

Please find attached a reply to the reviewer comments. We thank you and the anonymous reviewer for the thorough and constructive comments and appreciate the time and effort that you have invested in improving this manuscript.

Thank you again for a thorough and efficient review process. We will be glad to answer any further questions.

Sincerely,

Nimrod Wieler

**Comments by Reviewer**

**In the manuscript "Estimating the growth rate in desert biological rock crusts by integrating archaeological and geological records" by Wieler et al., the authors propose a creative method for dating the age of archeological sites by evaluating the growth rates of the rock-crust of biological origin. Although a helpful and required method, the assumptions and constraints made should be better specified.**

A: Thank you for this constructive feedback, please see our reply to specific comments below.

**R: It is not clear to me whether the studied rocks are defined as desert varnish rocks, or under a broader case of rock crust. On one hand, the authors reference varnished rock studies but do not term them as such. I'm afraid that in some cases, this may mislead the readers. For example, the paper by Lang-Yona et al. does indicate that fungi are scars in type I varnished rocks, but this does not mean that other rock-crusts (or other types of varnished rocks) will have the same microbial composition. Varnished rocks can be clustered into five categories, based on their elemental composition, formation rate, and structure, implying possible differences in their formation mechanisms (Macholdt et al., 2017).**

**Therefore, a more precise definition of the type of studied rock-crust is needed, in addition to accuracy in citing other types of crust structures.**

A: Thank you for pointing out this vagueness. The crusts we studied are hardpan-laminated structures composed of masses of micritic to microsparitic carbonate layers, interbedded with microbial coatings and, in turn, cover the lime and chalk host-rocks (as defined in lines 74-75). Such laminated structures were previously reported by Alonso-Zarza and Wright, (2010) as stage four terrestrial calcretes. Comparison of the microbial community between the different biological rock crusts is mentioned as a base for broader scale of understanding of such rock crusts growth rates. We use the term biological rock crust (BRC) as it was previously coined by Gorbushina (2007) for microbial communities that develop on solid mineral surfaces exposed to the atmosphere. We have clarified this in lines 22-25

**R: The authors estimate a general growth rate value for biological rock crust (BRC), based on the chalk and limestone crust thickness and the assumption of crust accumulation starts with the establishment of the site. While the idea is creative and provides a helpful tool for archeological dating, I am not convinced that this equation can be applied to other environments and different types of rock crusts for the following reasons:**

1. **As the results show, the thickness of the chalk crust is 1-fold smaller than that of the limestone (line 72-74), and the thickness is not even over the same rock, as in limestone. This indicates that either the rate of formation is not even over different locations, or that the crust degrades/falls off with time. In the first case, one cannot assume a constant rate of formation, and in the latter case, how can the actual thickness of the crust representing the zero-starting point of crust accumulation can be determined? I presume assumptions have been made here, but they should be clearly stated, in order for this tool to be applied.**

A: Thank you for mentioning it. The BRC thickness on the two lithologies (i.e., chalk and limestone) at the Byzantine site was measured and compared to the chalk and limestone BRC at natural rock outcrops located in the adjacent slopes. As a result, we find the chalk blocks at the Byzantine site as a reliable zero starting point for estimating BRC growth rates (lines 130-132). The thickness of the crust is indeed uneven, to some extent. This is to be expected and is precisely why our measurements provide a range of values for estimating the growth rate of the crusts, rather than a single point estimate.

2. **The type of stone is only one constrain. Others include the directionality of the stone and exposure to sun radiation, the porosity of the stone, mineral content, humidity, slope directionality, etc. The authors should constrain their proposed rates to hold safe under specific conditions, and as an average rate with possible upper and lower limits.**

A: Thank you for raising this issue. To avoid the slope aspect effect that may lead to different moisture regime, all samples were retrieved from a south-facing slopes/walls

and were collected during the same time period (lines 220-222). To constrain the geotechnical properties of the subjected lithologies we performed different analyses including porosity, rock bulk density and mineralogy (Table 1). Naturally, our growth estimates cannot be translated "as is" to any setting, but must be calibrated to match the local climate conditions and mineral composition. This is now clarified in lines 134-13523: "Calibrating of local climate conditions and mineral composition is needed for BRCs growth rates when applying on different conditions"

**3. I am missing a validation test for this method. Did the author sample other sites and tried calculating the age of the stone based on the thickness of the BRC?**

A: Thank you for the suggestion. We now conducted a validation to the BRC thickness dating method at a different archaeological site called " Nitzana Byzantine site". The new site is also located in the Negev Desert, Israel; some 20 km from our original site. Our measured BRC thickness results at this site show a 100-200 μm crusts. This is now clarified in the "Method validation" section (lines 84-88) and in Fig. S2. These findings are consistent with the BRC thickness crusts that were measured at Shivta archaeological site.

**R: The statement of bio-crust causing the difference in $d^{13}C$ between the rock types is thin. The authors do not present analyses of negative control rocks with no bio-crust, to prove that the difference in $d^{13}C$ values between chalk and limestone indeed comes from the crust activity. Therefore, I am not convinced that this is the reason for the difference, rather than the age of the rock, the structure, porosity, density, etc.**

A: Thank you. To confirm the biogenic nature of the BRC upon the two lithologies we conducted isotopic cross section for each lithology separately. This analysis was conducted for BRC's both from the Byzantine site and the adjacent natural slopes. The $d^{13}C$ profiles compared the crust to their host rock that underlie beneath it. The host rock was used as the negative control for each lithology in the given cross section. We find the limestone BRC $d^{13}C$ values to show large difference compared to the host rock (lines 91-93, Fig.3A). No similar clear observation was found upon the chalk BRC $d^{13}C$ values (lines 93-94, Fig.3B). The differences in $d^{13}C$ values between the chalk and limestone are suggested to result from the BRC thickness. Thicker BRCs, as observed in the limestone samples, may hold more biogenic activity compared to the chalk (lines 97-98).

**R: In figure 4C, how does the stone type impact the coordinates of the samples' bacterial composition? This would be a valuable addition of information into the analysis. In addition, PCoA or other such analysis linking different parameters to the microbial composition may also be a valuable addition. For example, the impact of porosity, surface-to-volume ratio of the crust, crust thickness, and other parameters on the distribution of community composition of the samples in the coordinate matrix may give a hint on key microbes' preference under different conditions.**

A: As can be seen in Table S2, variance-paritioning analysis using PERMANOVA showed neither a significant effect of the lithology nor of the interaction between the lithology and sample source on the community compositions. Hence this was excluded from the model shown in Fig. 4. This similarity in composition demonstrates the indifference of microorganisms to the type of attachment surface in this case and that the community probably changes very little after establishing. However, the differences in BRC thickness between the chalk and limestone sampled from the slopes could indicate that the latter can better support BRC growth (lines 138-140).

---

## Author Comment (AC2)

Dear Editor,

Please find attached a reply to the reviewer comments. We thank you and the anonymous reviewer for the thorough and constructive comments and appreciate the time and effort that you have invested in improving this manuscript.

Thank you again for a thorough and efficient review process. We will be glad to answer any further questions.

Sincerely,

Nimrod Wieler

**Comments by Reviewer**

**Manuscript: „Estimating the growth rate in desert biological rock crust by integrating archaeological and geological records by Weiler et al. is valuable study of biocrust concerning its origin, taxonomy, isotopic composition, and growth rate. Bellow there are some comment.**

 A: Thank you, please see our reply to specific comments below.

**Major comments**

**R: Manuscript title focuses solely on „growth rate". Large portion of manuscript is nevertheless focused on bacterial communities based on RNA sequencing and also on isotopic composition of rock and crust. These two topics should be also mentioned in paper title. Paper title should be changed, for instance: „Desert biological rock crust: Bacterial diversity, isotopic composition and growth rate". This might better express manuscript content.**

A: Thank you for the comment. We believe that the focus of the manuscript is estimating the biological rock crust growth rate. Especially, due to the limited amount of studies dealing with this issue in arid environments. We agree that there is also a lot of focus on the isotopic composition and microbial community and have therefore changed the title to:

"Microbial and geo-archaeological records reveal growth rate, origin and composition of desert rock surface communities"

**R: Also abstract should better correspond to manuscript content. There is not a single sentence on isotopes for instance.**

A: Thank you. The abstract was revised so that it also refers to the isotopic analysis.

**R: Conclusion is very short and not much reflecting findings of the manuscript. It should be rewritten to cover all important findings of the paper.**

A: Thank you. The conclusion section has been rewritten so that it includes our major findings.

**R: As one of the most important results of the paper is the growth rate of biological rock crust, it is important to describe on how many samples the thickness 0.1 -0.6 mm was measured.**

A: Thank you. To estimate biological rock crust growth rate we used total of 24 rock samples. Twelve rock samples were collected from the limey Nezer Formation including six samples from the archaeological site and six from the nearby natural slope. The same sampling procedure was applied for the chalky Menuha Formation (lines 217-222). This is now clarified in lines 72-73: "Microscopic examinations of thin sections (30 μm thick) that were prepared from 24 limestone and chalk blocks showed that, as expected, the BRCs were restricted to the atmospherically-exposed parts of the rocks (Wieler et al., 2019)".

**R: Line 149: Authors state that: „Our analyses invalidate the lithology role in shaping the BRC composition". In fact not really, as limestone and chalk are both calcite rich, so from many respect very similar lithologies. Statement is too strong, not supported by data. Statement should be changed or more evidence is needed.**

A: We appreciate the comment and agree with the reviewer that chalk and limestone are very similar lithologies as can be seen through the XRD analysis (Table 1). Following the reviewer comment the statement was modified (line 163) - "Our analyses suggests that both rocks experience the same regional-scale environmental factors (Fig. 4B) in shaping the BRC composition."

**Minor comments**

**R: Fig. 1 Colors in figure are so dull that individual lithologies can only hardly be recognized. Colors should be brighter. Also, what means the red color, which lithology?**

A: The figure was modified following the comments above. It further includes a definition of the other lithologies projected on the map.

**R: Lines 80-90: Concerning O a C isotopic values, the name of standard should follow after values (like -3‰ VPDB)**

A: Correction of the O and C isotopic standard was added along the subjected lines.

**R: Fig. 3 The description of horizontal axis "d13C/a18O" may confuse that ratio of C/O isotopes is used. Better would be: "d13C and d18O". It would be nice to separate in graphs which part of the data are in crust and which are measured below the crust (for instance using black dashed line as base of rock crust or highlight the rock crust by grey color)**

A: The figure was modified following the suggestions raised by the reviewer.

**R: Fig. 4 Figures A-D have small fonts, especially the B which is way too small.**

**Better alignment of individual parts of figure will be**

**A C**

**B**

**D**

**So that B a D would have whole width of page.**

A: The figure was modified following the suggestion raised by the reviewer.

**R: Line 130 "Overall, the results point to a very deterministic successional course of BRCs development." Meaning of sentence is not clear, please rewrite.**

A: The line was rewritten (line 143): "Therefore, our findings indicate that the rock surfaces at the archaeological site act as a succession planes for BRC development."

**Line 223-225. It seems that O isotopes are in fact referenced with respect to VPDB rather than VSMOW. Otherwise, O isotopic value of marine limestone would not be close to 0. Anyway, this should be clear from manuscript.**

A:The lines (240-242) were rewritten: "d13C and d18O values were also referenced relative to Vienna PeeDee Belemenite (VPDB) standard as previously described (Uemura et al., 2016) with SD of 0.1‰. All values are reported in per-mil (‰)."

---

## Author Comment (AC3)

Dear Editor,

Please find attached a reply to the community comment. We thank you and the large group of people for the thorough and constructive comments and appreciate the time and effort that you have invested in improving this manuscript.

Thank you again for a thorough and efficient review process. We will be glad to answer any further questions.

Sincerely,

Nimrod Wieler

**Comments raised by the community:**

**This review was compiled by a senior undergraduate class in critical thinking in ecology and environmental sciences, with the purpose of understanding and contributing to the peer review process. We hope our comments will help the authors improve their manuscript.**

**Kate Buckeridge, Keshia Ashaari, Boyan Karabaliev, Francisco Navarro Rosales, Lea Opitz, Beth Simms, and Daphne Ziogas**

A: Thank you for this constructive feedback, please see our reply to specific comments below.

**Re: the main question addressed by the research**

**The main purpose of this research was to make the first ever quantitative estimation of the growth rate of biological rock crusts (BRCs) in arid regions under natural conditions. The researchers hoped that the findings would enable BRC thickness measurements to become an affordable method to date archaeological sites in arid environments in the future.**

BRCs are relative in a small number of areas of biogeochemistry; they determine the soiling and weathering of rocks in arid environments and are thought to have played an important role in the structuring of dry lands. Additionally, they cause damage to important archaeological heritage sites over long periods of time. They are also relevant to extra-terrestrial research in that they may provide an analogue for possible life on Mars. Whilst there are a few areas of research where biological rock crust growth rate is relevant, the value or usefulness of its application to these areas is not expressed in the manuscript. This lack of further explanation regarding the application of the findings of this study reduces interest to the reader.

A: Thank you for the comment. We further defined the possible application of our study, This is now clarified in lines 36-37: "Illustrating the origin, composition and growth rates of BRCs may shed light on the timescales where natural processes (e.g., desertification, land degradation) take place in arid regions."

**Discrepancy between title and RQs: title suggests the ms (title, introduction, discussion) is about growth rate, but other sections (methods, results) revolve around the use of isotopes and microbial community, without relating how these methods will help to understand growth rate. Please be more expansive in title and objective, to describe what is a BRC.**

A: Thank you for the comment. We believe that the focus of the manuscript is estimating the biological rock crust growth rate. Especially, due to the limited amount of studies dealing with this issue in arid environments. We agree that there is also a lot of focus on the isotopic composition and microbial community and have therefore changed the title to:

"Microbial and geo-archaeological records reveal growth rate, origin and composition of desert rock surface communities"

**Re: the subject area and comparison to other published material**

**The article claims that it presents the first estimation of the growth rate of BRCs in an arid environment. However, the object of the study is somewhat unclear as the term "biological rock crust" is not defined in any of the articles cited (perhaps l. 68-69?). For example, "biological rock crusts" refers to both the live biological layer and the accumulation of abiotic material (Table 1), the latter being the studied phenomenon.**

**The term "biological rock crust" is not widely used. In fact, the only other mention of that term appears to be Wieler et al (2019). (Scopus search: TITLE-ABS-KEY ( {biological rock crust}  OR  {biological rock crusts}) ). Gorbushina (2007) is cited under the definition (line 20) but does not define or mention "biological rock crusts". (Neither do Lebre et al., 2017; Pointing and Belnap, 2012; Lang-Yona et al., 2018, all cited in the introduction.)**

**The term "biological soil crust" (BSC) is commonly used, including for BSCs growing on a mineral subtrate. The term "biological soil crust" does not appear in the text. We strongly encourage the use of an already established term, e.g. Pointing and Belnap (2012).**

A: Thank you for pointing at this issue. As the term Soil Rock Surface Communities (SRSC), mentioned by Pointing and Belnap (2012), refers to a broad scale of mineral substrates (soil and rock surfaces), we found it important in our work to differentiate between the general term and the rock communities. Therefore, we suggest using the term Biological Rock Crust (BRC) as it reflects better the phenomenon.

**It is unclear why the abstract and introduction claim BRC growth rates have never been quantified (Line 31: "Currently, there is no available information about the development rates of epi-and endolithic lithic biofilms on rock surfaces in deserts."), yet in lines 109-115 a few examples of estimation are given. The difference between this study Lange (1990) (cited line 111) appears to be that the cited study quantified lichen rather than microbiotic crusts; the article should make that distinction clear. However, in lines 112-115, two studies quantifying the growth rate of desert varnishes (which are hypothesised to be of microbial origin) are cited as examples. The authors imply that desert varnish is a type of BRC (lines 22-23, lines 110-112), however, the biotic origins of desert varnish are controversial: for example, Lang-Yona et al. (2018) (which was cited in the introduction, lines 22-23) suggest that the main processes behind desert varnish formation aren't biotic, though microorganism traces are present, quantified via genomic sequencing similarly to this article.**

A: We agree with the reviewers that the examples given (lichen growth, desert varnish) may arise questions when comparing to BRC growth, yet while going over the existing literature, these were the only possible references we found that should be compared to. Moreover, we rephrased the text so that BRC can be easily distinguished from the other existing rock crust, as mentioned now in lines 23-25- laminated structure composed of masses of micritic to microsparitic carbonate layers interbedded with microbial coatings covering the lime and chalk host-rocks.

**Re: Structure, style, and presentation**

**This paper follows a formal scientific style and structure which helps the text flow. It would be clearer if the methods section is presented after the introduction. The introduction sets the context of the study, however, although some questions are mentioned, it is unclear what the hypotheses of the study are, which negatively affects the whole paper. One goal is presented (growth rate estimate) but further hypotheses (linked to methods and results) would tie together the manuscript story. The results of the study are clearly presented. The discussion is carefully written stating claims and referring to the data, although, as mentioned previously, some points seem to be overrepresented (i.e. stable isotopes, microbial community) because they have not been mechanistically tied to the research question. This is because the hypotheses of the study have not been stated clearly in the introduction. Rephrasing of the questions to be addressed in this study will give the paper clarity. The methods can be more reliable if it is stated how each methodology used answers the hypotheses and if more detail is used.**

A: Thank you for the positive feedback. Following the reviewers comment we rephrased the hypothesis in the introduction section.

**The writing is clear and succinct. The abstract summarizes the key findings and scope of the study effectively. The text throughout is precise and frequently referenced. However, knowledge often seems to be assumed. This again can be avoided with the use of clear objectives and rephrasing of the title. There are a few minor grammatical errors (lines 58, 131, 132).**

A: Grammatical errors were corrected and the title was changed.

**Section numbering is incorrect: should be 1.0 Introduction; 2.0 Results, 2.1 Geotechnical properties, etc**

A: Section numbering was corrected

**Re: the conclusions and consistence with the evidence and arguments presented**

**The main conclusion of the study is that the growth rate of biological rock crusts has been estimated as 0.06-0.35 mm Kyr-1. These results directly fulfil the study's main goal (to provide a first quantitative estimation to the growth rate of lithic**

**communities on rock surfaces in arid regions under natural conditions; line 34). They are important results which are highlighted in the abstract, results, discussion and conclusion sections. The authors are able to clearly transmit their main finding to the readers.**

**However, the way by which the BRC growth rate was calculated is not entirely clear, and the author's conclusions do not seem to be entirely consistent with the evidence provided. The authors do not explicitly explain how the growth rate was calculated. I assume the authors have calculated the BRC growth rate by dividing the (0.1-0.6 mm) thickness values by the age of the site (1700 years). This calculation produces the expected growth rate of 0.06-0.35 mm Kyr-1. If this is the case, the authors could just easily explain their calculations in one or two lines.**

A: Growth rate was calculated by dividing the BRC thickness by the age of the site. This is now clarified in the thickness measurement section. Validation to this method was further applied in another archaeological site in the Negev Desert (i.e., Nitzana) as presented in the method validation section.

**Still, the growth rate calculations show two further inconsistencies. The first one is that the authors have only used the earlier/older age bound whilst calculating rate. The authors clearly state the Byzantine site of Shitva is dated between the 4th and 7th centuries (line 75), meaning buildings could have been built 1700-1400 years ago. Having an age range of 300 years introduces a significant source of uncertainty and limits the strength of their conclusion. Perhaps, the authors could justify why they have decided to stick with the older bound of the archaeological range.**

A: We agree with the reviewer that the buildings at the site introduce a significant source of uncertainty. We could not determine at this point of the research the exact age of each wall, and as a result we prefer to take the conservative approach and stick to the older bound.

**The second one is that the authors just calculated BRC growth rate using thickness values in chalk. At first glance, this looks rather illogical. Why would the authors use values from just one rock type? Having re-read the manuscript multiple times, I can now see the authors just calculated growth rates for BRCs in chalk because limestone BRCs were not removed at the time of construction (lines 57-58). This makes sense, growth rates can only be calculated if BRCs grow on bare rock. The authors are being misleading, because they have never clearly stated that growth rate is only calculated in chalk. They have talked about BRC growth rates in arid environments, implying they have studied both chalk and limestone type of rock.**

A: We refer our dating method to limestone and chalk as they are dominant rock types in the Negev desert and in deserts in general, This is now clarified in lines 48-49.

**This last point also limits the strength of another of the authors' conclusion, that atmospherically exposed archaeological artifacts could be dated by measuring the**

thickness of BRCs with known regional growth rates (lines 166-168). The study has found that BRC thickness varies between different rock types (line 121), meaning different growth rates will be supported upon limestone and chalk (lines 126-127). Thus, it is likely archaeologists would need rock-specific growth rates to calculate age. Additionally, BRC growth rates are also likely to vary along environmental gradients, since the authors have only retrieved samples from south facing slopes to remove any variation from differences in moisture regime (lines 203-204). The evidence suggests growth rates will be too variable and uncertain for scientists to accurately date archaeological sites. I understand the authors want to highlight the relevance of their research by presenting a novel dating tool, but it is too much of a bold statement at this early stage of research.

 A: We agree with the reviewers that the suggested growth rate is limited to specific environmental conditions. The growth rate findings at both archaeological sites suggest the possible applicability of such dating method when located in arid regions, and further research should be conducted when applied in other regions.

The other study conclusions are included within the discussion section. They are generally well argued, clear and easy to understand. But perhaps the authors could further explain some of their statements or include some additional evidence to support their argument. For example, in line 125, the authors suggest similarity in chalk and limestone BRC composition demonstrates microorganisms are indifferent to the type of the attachment surface and that BRC communities change very little after establishing. I can easily understand their first claim but would appreciate some more evidence that supports their argument of little community change. Is it possible BRC communities could have only become compositionally similar in the last 100 years out of 1700? Similarly, in lines 131-132, the authors state that the similarity in community composition between samples close to the ground and away from the ground indicate a major role of aeolian processes in determining the BRC composition. What aeolian processes are they referring to? Does it refer to dust particles, which can be blown by wind, being the main potential source for the microbial communities (lines 151-152)?

 A: Stating that BRC communities changes very little after establishing is based on the minor differences that were observed between the communities in the site vs the communities in the natural slope. As the natural slope records longer term colonization and the site records short term we suggest that along this timeline the community does not change dramatically. As for the aeolian processes, we refer to settled dust particles; this was added to the text, following the reviewer comment.

Overall, the authors' conclusions regarding which variables influence BRC composition are coherent and consistent with the evidence. Community composition is affected by aeolian processes and regional environments but not by lithology or by proximity to the ground. Still, the authors should remember to clearly acknowledge that their data is limited to a few locations, and that the patterns they observed may not occur in other regions, or in other locations within the Negev Desert. The

authors compare their results to plenty of other research about BRC-like system, which show different ecological patterns and highlight the issue of limited replication in location. It is clear the authors are aware of the current state of science surrounding biological rock crusts. The information they include is rather helpful, although the way in which they present it is not ideal (it is quite unclear why the authors decide to talk about desert varnish in their first discussion paragraph). They are able to use findings of other studies to support their conclusions regarding the importance of different bacterial taxa within the BRC community composition (lines 135-147).

A: Thank you for this feedback. Desert varnish was compared to the BRC microbial community as it is a common crust upon many rock surfaces in arid regions. Although its formation is under debate we find it relevant to compare it to our results.

**Re: Tables or figures**

The authors did a good job in constructing the figures and tables. Figures are referred to in the main text and appeared in the order of numbers. The captions are detailed and mostly explain the figures well without the need for the reader to refer to the main text. At first glance, the figures and tables capture the reader's attention and highlight relevant sections of the paper. They hold a well-designed, professional appearance which entices the reader to read through the figure captions and full manuscript.

Figure 1 is a good introductory figure which illustrates the study site. The reader is unlikely to be familiar with the study area and it was difficult to imagine the area solely based on its description in the main text. This figure supplements the text by clearly showing the geographical location of the study area with a good view of the Shivta Byzantine city and its adjacent slopes.

A: Thank you.

The authors presented a clear contrast in BRC thickness between the chalk and limestone blocks from the city and slopes in Figure 2. In figure 2A we can clearly see the difference between the chalk blocks and the limestone blocks although the solid and dashed arrows are quite inconspicuous. From Figures 2B and 2C, the yellow bars emphasize the difference in chalk BRC thickness between the city (B) and adjacent slopes (C). They also help the readers locate the BRC within the figure. However, the thickness measurements in red are hardly noticeable and perhaps unnecessary as this information is already available in the main text. Although the caption mentions that the dashed lines mark the border between the BRC and the host rock, the authors did not indicate the location of these components: whether the BRC is above or below the dashed lines. Since the yellow bars were used for Figures 2B and 2C, it may be useful for the authors to include them in 2D and 2E as well.

A: Thank you the figure was modified following the comments.

In Figure 3, we can clearly see how the δ13C and δ18O values change with depth (from BRC to host rock) in the city and slope samples. At first glance, this figure seems to have delivered its main message well. However, upon further review, there are a few things that remain unclear. In the main text (lines 79-80) the authors mentioned the depth of the limestone BRC layer (0-2 mm) and host rock (2-5mm). Is this the same for chalk samples? If not, why was this not stated in the results section? We do not know the thickness of the BRC and host rock from the figure and its caption. The depth of the layers was not illustrated in Figure 3 even though the caption clearly states, "Carbon and oxygen isotopes profiles in BRCs and host rock samples of limestone…". How would the reader know at what depths the crust and host rock are located? Perhaps overlay a line from the y-axes at the BRC-host rock interface.

The authors also stated that the results were consistent for both the slope and archaeological site samples, with the limestone δ13C values for the BRC ranging between -4‰ and -5‰ and host-rock values between 0‰ and 1‰ (lines 80-82). However, when referring to the figure only, it is only the pattern of isotopic composition change that is consistent, not the value itself. The host-rock values mentioned in the main text are true for the city samples but based on Figure 3, the slope samples had δ13C values of -2.0‰ or less. It was also stated that the host-rock's δ13C values for the chalk ranged between 0.1‰ and–2.7‰ (line 81), but the maximum value for the x-axis in Figure 3B was only 0.0‰.

A: Thank you, an overlay line is added in the figure to distinguish between the BRC and the host rock. The isotopic values presented in the figure indicate the trend we observed in several isotopic profiles we carried out along the chalk and limestone samples. That may explain the discrepancy between the trend in the figure and the exact values presented in the text.

Without referring to the main text, Figure 4 clearly communicates the main results of bacterial diversity of the BRCs in the city and adjacent slopes. It is observed that regardless of lithology or BRC thickness, all samples demonstrated very similar microbial community composition. It is also obvious that Actinobacteria is the most abundant phylum in chalk and limestone samples in both the city and slopes. A minor point the authors missed is the inconsistency in colour chosen to represent the slope and city data. The fonts are also very small and difficult to read. In Figure 4A the orange plots were used for slope samples and green for city samples, but the opposite was done for Figure 4D. To avoid confusion the colours could have been standardised for all subfigures.

A: Thank you, the figure was modified following the reviewers comments.

Table 1 summarised the geotechnical properties of the subjected lithologies effectively and concisely. Overall, the figures and tables presented by the authors supplement the main text well, but there are some inconsistencies between the text

**and figures. The supplementary figures and tables were also labelled inconsistently (i.e. Table S1 in the main text but Table A1 in the appendix).**

**Re: methods and results**

**The method section is sufficiently detailed. A sample size of six is adequate and the number of replications for certain analyses is mentioned throughout. The standards used for the calibration of chemical analysis are named which simplifies possible repetitions (lines 222-225). The description of the R packages used was detailed (Section 1.4.5.) and making the code as well as data available increases the transparency and reproducibility of the statistical analysis. Overall, the methods are suitable to answering the research questions regarding the growth rate (in thickness) but it is unclear where some of the other methods come into play.**

A: Thank you, we clarified the potential contribution of each of the methods to the origin and composition of the BRCs.

**The results were phrased in a concise and clear way and reporting findings in ranges does helps to communicate the uncertainty associated with them well. There are however some structural issues (assuming methods follow introduction): point 1.1.1. is more a description of the study site than a result so might better be incorporated in the study site section of the methods. In section 1.1.3. (lines 85-88) some results are already being interpreted, which would better fit in the discussion section (or for l. 85-86, in the introduction, to explain why this method is being used). Parts of the methods and results are not found in the research questions which could be addressed by either including methods like DNA extraction (section 1.4.4.) or isotopes (section 1.4.3.) in the introduction and then in the discussion but in relation to the research question. If these analyses do not add to the main finding of the paper it could be worth not mentioning them.**

A: Thank you. We believe the geotechnical properties section should be incorporated in the result section as it sets the geological context of the studied BRC composition. The introduction section was modified to better link the research question to the results and discussion sections.

---

## Author Response (AR1)

Dear Dr Akob,

Please find attached a reply to the comments. We thank you and the anonymous reviewers for the thorough and constructive comments and appreciate the time and effort that you have invested in improving this manuscript.

Thank you again for a thorough and efficient review process. We will be glad to answer any further questions, please see our reply to specific comments below.

Sincerely,

Nimrod Wieler

Comments

**1. Please modify the organization of the paper to present the materials and methods before the results section. I agree with the comments from Buckeridge et al that this order would be more effective for understanding the study and approach. Further, this structure is more consistent with others papers in Biogeosciences.**

A: The structure has been revised.

**2. L. 13: please correct the spelling of archaeological and comparable**

A: Spelling correction was made.

**3. L. 26: please capitalize taxa names when referring to the formal taxonomic designation**

A: Taxa names are now capitalized

**4. Figure 2: I find it extremely hard to reach the text on panels B, C, D, and E. The scale bars are not apparent in the PDF and the text is too small for the units. Further, the red text for the BRC thickness is not readable. One option might be to reproduce the panels as a full page in the supplemental material so that the reader can see the images enlarged. Or maybe the labels/text need a white box behind them.**

A: The figure was modified following your comments.

**5. L. 84 and elsewhere: it's a bit unclear to me (and the reviewers) why a range of calculations are not presented. I also don't agree with your response that choosing an older age is a more conservative approach. I think adding a few sentences to present calculation of the growth rate for both 1700 and 1400 years ago would be valuable and not outside the scope of the project. Also, make the uncertainty in the site age clear – folks reading this paper might not be familiar with the uncertainty in archaeological dating.**

A: Dating Shivta site is mainly associated with the early-middle Byzantine period (4th-6th centuries CE) (1700-1500 years ago) as was recently validated by Tepper et al., (2018). As a result, dividing our observed BRC thickness over the site age suggests BRC growth rate of 0.06-0.35 mm 1000 yr-1. We modified the thickness measurements paragraph to further clarify this point.

**6. Figure 3: please define the dotted line in the figure description. I also suggest using the same axis range for panel A and B.**

A: The dashed lines mark the border between the BRC at the top and the host rock at the bottom. This is now added to the figure caption.

**7. Section 1.2.2: I'd really like to know more about who the dominant OTUs are or even just the most abundant OTUs that had significant changes in abundance (e.g., corncob). Right now the data is being presented at a really high taxonomic level and nothing is stated about the potential life strategies for these organisms. It would be interesting to know on a lower taxonomic level if there are typical arid soil organisms or super tolerant species. Maybe even going to just a family level would be sufficient.**

A: We have now added a brief discussion about the identity of the top OTUs in lines 227-232 and lines 262-267 and an addional supplementary table. These OTUs turned out to be typical for hyperarid desert soils. However, we kept the discussion here relatively short because our anaysis indicates that which exact OTU is dominant is strongly affected by stochastic processes (founder effect).

**8. L. 141-142: is this meant to be a standalone paragraph? Seems like this should be part of the paragraph above.**

A: The lines were added to the paragraph above.

**9. L. 151: similarity**

A: The line was added to the paragraph.

**10. L. 153: can aeolian processes be introduced earlier in the paper? Expanding on this process and its influence would be helpful**

A: The possible impact of aeolian processes on BRC formation is further added to the introduction section to better suggest the possible matrices (soil, dust) which may lead to BRC formation.

**11. L. 183: use BRC here to be consistent with the rest of the paper**

A: Correction has been made.

**12. L. 184: this value differs from that presented in the abstract. The range should be used in the abstract for consistency.**

A: Similar range of values are now presented both at the abstract and at the conclusion sections.

**13. Section 1.4: I still find the conclusions section to be short. I think it would be valuable to broaden the discussion here and talk about the broader applications of the study.**

A: The conclusion section was modified so that it further discuss the possible origin of the BRC.

**14. Section 1.5.1: its unclear to me what the actual age of the buildings are. If there is a range, please state that explicitly so its clear to the reader.**

A: Shivta site was found to be most prosperous in the early-middle Byzantine period (4th-6th centuries CE) (1700-1500 years ago) as was recently validated by Tepper et al., (2018). Therefore, most archeologists agree that this is the building age. This is now clarified in lines 209-210.

**15. Table 1: can the footnotes be included as an entry in the table? They are data that are important to the study. Correct the superscript for host rock density units**

A: The footnotes (geological formation names) were added into the table, the superscript was corrected.

**16. Sampling: can more detail on how the samples were collected be included? Were whole rocks removed or rock cores collected then subsampled in the lab?**

A: All rock samples taken from the slopes and site walls included whole rocks that were collected directly in the field and were not subsampled in the lab. This now added to the field sampling section.

**17. L. 280: correct spelling to indices**

A: Correction has been made

**18. L. 282: do you mean package?**

A: Correction has been made

**19. L. 284: I can't find any mention of an "non-parametric igned Rank Transformation ANOVA" – is this a typo?**

A: This should be "non-parametric Aligned Rank Transformation ANOVA" (now corrected) and refers to Figure 4 D.

**20. Figure A1: since this is a supplemental figure please make it bigger to take up as much of the page as possible. This way the detail is really easy to see**

A: The figure was enlarged.